# The Role of Quantitative Real-Time PCR in the Invasive Pulmonary Aspergillosis Diagnosis: A Retrospective Study

**DOI:** 10.3390/microorganisms13020409

**Published:** 2025-02-13

**Authors:** Laura Trovato, Maddalena Calvo, Concetta Ilenia Palermo, Guido Scalia

**Affiliations:** 1Department of Biomedical and Biotechnological Sciences, University of Catania, 95123 Catania, Italy; lido@unict.it; 2U.O.C. Laboratory Analysis Unit, A.O.U. Policlinico “G. Rodolico-San Marco” Catania, 95123 Catania, Italy; maddalenacalvo@gmail.com (M.C.); ci.palermo@policlinico.unict.it (C.I.P.)

**Keywords:** invasive pulmonary aspergillosis, real-time PCR, *Aspergillus* spp., respiratory samples

## Abstract

Invasive pulmonary aspergillosis (IPA) reports significant mortality rates among critically ill patients. A prompt microbiological diagnosis is essential to establish a coherent antifungal treatment. Despite its low sensitivity and prolonged turn-around time, culture represents the conventional diagnostic technique. Additionally, galactomannan detection may support the diagnostic process. Ultimate generation methods, such as the real-time polymerase chain reaction (Real-Time PCR), integrated the diagnostic procedure to improve the overall laboratory effectiveness, especially regarding a quantitative *Aspergillus* spp. DNA detection. Herein, we propose a retrospective analysis where a quantitative real-time PCR was performed on respiratory samples belonging to patients with or without probable pulmonary aspergillosis. The study enrolled 62 samples, whose PCR results were compared to culture and galactomannan indexes. Additionally, clinical and general data were collected for all the patients. The qPCR assay reported 100% sensitivity and negative predictive value, while specificity reached 59.2% and the positive predictive value was 76.1%. Moreover, IPA patients reported fungal DNA loads higher than 10^3^ in a logarithmic scale, while non-aspergillosis episodes reported a maximum level of 10^3^. We hypothesized a future possibility to define a specific cut-off in distinguishing colonization from infection cases, requiring further investigations and speculations about IPA patients and respiratory samples.

## 1. Introduction

Invasive pulmonary aspergillosis (IPA) represents a significant public health threat, accounting for increased mortality rates and a global incidence higher than 300,000 cases among approximately 10 million patients carrying specific risk factors. Several clinical trials documented the highest invasive aspergillosis incidence (>85%) in immunocompromised patients, subjects with chronic respiratory illnesses, and transplanted patients. Specifically, lymphoproliferative disorders, prolonged and significant neutropenia, oncological diseases, and solid organ transplantation predispose patients to invasive fungal infections. Furthermore, intensive care unit (ICU) patients often suffer from prolonged corticosteroid usage, cardiovascular diseases, and respiratory illnesses, which may be underlying conditions for IPA development [1,2]. Invasive aspergillosis recently emerged as a complication for patients previously affected by viral pneumonia, especially within the ICU setting. According to a scientific report [3], more than 50% of these patients did not carry specific risk factors except mechanical ventilation, neutrophil dysfunction, and immunosuppression due to viral infection. SARS-CoV-2 and *Orthomyxoviridae* are the most important predisposing viral agents for IPA occurrence [1,3]. Medical guidelines identified influenza-associated pulmonary aspergillosis (IAPA) and COVID-19-associated pulmonary aspergillosis (CAPA) as specific clinical conditions when *Aspergillus* spp. infections emerge after viral respiratory aetiologies among ICU patients [3].

All the clinical conditions related to invasive aspergillosis need an early microbiological diagnosis, requiring a consequent targeted antifungal treatment. Culture-based and microscopic techniques represent a conventional method in aspergillosis diagnosis. Moreover, cytopathologic and histopathologic hyphae presence on tissues confirm this infection [4]. Serological markers such as β-1,3-D-glucan (BDG) or galactomannan (GM) may support the invasive aspergillosis laboratory diagnosis [4]. Unfortunately, all the conventional diagnostic methods require prolonged turn-around time (TAT) and suffer from low sensitivity rates.

Different molecular methods detect *Aspergillus* spp. DNA on various biological samples. Additionally, some kits specify potential resistance markers or quantitative information [5]. For instance, some in-house or commercial real-time PCR protocols enabled *Aspergillus flavus*, *Aspergillus terreus*, *Aspergillus niger*, and *Aspergillus fumigatus* identification. Moreover, these molecular kits allow cyp51A mutation research, especially regarding the common L98H and TR34 related to *A. fumigatus* azole resistance [6]. All the consolidated molecular methods base their analysis on two main targets, which are 18S rRNA (whole blood samples) or mitochondrial DNA (serum samples) [7]. Previously published data demonstrated a similar PCR performance for serum and whole blood samples, documenting the importance of achieving at least two positive PCR test results to increase specificity in invasive infection diagnosis [7]. Consequently, the double-positive result collection became an official European *Aspergillus* PCR initiative (EAPCRI) recommendation for all the clinical laboratories performing PCR tests in IPA diagnosis [7,8].

Despite its previous exclusion due to discrepancies in clinicians’ opinions, the European Organization for Research and Treatment of Cancer and the Mycoses Study Group Education and Research Consortium (EORTC) guidelines recently included PCR techniques for IPA diagnosis [9,10,11]. However, molecular methods are highly sensitive and may be positive without a real infection’s report. Some biological samples may reveal contamination by airborne conidia or non-specifically *Aspergillus* genomics fragments. Consequently, a fungal load specification could support clinical aspergillosis forms diagnosis, distinguishing them from non-aspergillosis infections, colonizations or non-invasive aspergillosis episodes [12]. Scientific literature demonstrated that fungal load expressed in Log10 was significantly higher in bronchoalveolar lavage samples from IPA patients than in patients without aspergillosis. These data inspired our study, aiming to support the *Aspergillus* spp. molecular detection and quantification in respiratory samples to differentiate between invasive clinical conditions and non-clinically relevant colonization cases. In other words, our analysis aimed to prove the effectiveness of a quantitative real-time PCR (qPCR) in invasive pulmonary aspergillosis diagnosis.

## 2. Materials and Methods

We performed a retrospective analysis including 62 patients from the University Hospital Policlinico of Catania, Italy. The study analyzed microbiological data (November 2022–2024) and included sputum and bronchial aspirate samples with at least a positive culture result for *Aspergillus* spp. On the other hand, the study involved bronchoalveolar lavages (BAL) with at least a positive culture result for *Aspergillus* spp. or a positive galactomannan detection (index value equal to or higher than 0.5). A datasheet collected all the general data, therapeutic information, clinical conditions, imaging results, and microbiological reports. The investigation retrospectively classified patients with probable invasive pulmonary aspergillosis following the EORTC criteria [9,10,11]. Probable invasive fungal disease diagnosis requires the presence of a host risk factor (neutropenia, transplants, immunosuppression), a clinical criterion (halo sign, air crescent sign, dense well-circumscribed lesions, lobar consolidation), and mycological evidence (microscopical detection, positive culture, galactomannan antigen detected in plasma or serum or bronchoalveolar lavage with the minimum index value of 1, duplicate positive molecular results on serum or plasma or bronchoalveolar lavage or whole blood) [10]. Patients reporting at least one host factor, one clinical criterion, and one form of mycological evidence were classified as IPA patients. Otherwise, patients showing negative records for all these criteria were classified as non-aspergillosis cases (non-IPA patients).

All the respiratory samples were restored from a −80 °C storage procedure, where their consistencies and properties were preserved after the routinary diagnostic protocols. After their selection, they underwent a qPCR test through the *Aspergillus* spp. ELITe MGB^®^ Kit (ELITechGroup S.p.A., Torino, Italy). This kit provided all the necessary reagents to perform completely automated extraction and amplification processes on the ELITe InGenius^®^ (ELITechGroup S.p.A., Torino, Italy). After the automated protocols, the system furnishes *Aspergillus* spp. DNA detection along with its quantitative copies/mL load within respiratory samples. The applied method showed a limit of detection (LOD) of 120 copies/mL, distinguishing between definitely negative samples (0 copies/mL) and DNA detection below the LOD (quantities included in a 1–119 copies/mL range). The qPCR result was registered into the datasheet, comparing its interpretation to culture-based (Vakutest Kima, Arzergrande, Italy and MALDI Biotyper^®^ Sirius System, Bruker, Billerica, MA, USA) method reports, patients’ clinical condition, and galactomannan (Platelia *Aspergillus* Ag, Biorad, Hercules, CA, USA) dosage.

The statistical association of galactomannan, molecular results, and culture reports to the aspergillosis or non-aspergillosis patients was reported. Furthermore, the authors calculated a statistical significance for patients’ clinical or general features such as antimicrobial treatment, sex, and age. We performed this analysis using the MedCalc Statistical Software version 17.9.2 (MedCalc Software bvba, Ostend, Belgium; http://www.medcalc.org; 2017; accessed on 13 December 2024), reporting the corresponding *p* values. Specifically, the χ^2^ and Fisher’s exact test were applied to establish the categorical variables as percentages. Medians with ranges were used to describe non-normally distributed continuous variables and compared using the Mann–Whitney U-test. The same software allowed the reporting of sensitivity (SS), specificity (SP), positive predictive value (PPV), and negative predictive value (NPV) percentages, along with the corresponding 95% confidence intervals.

## 3. Results

First, we observed the patients’ distribution within the analyzed hospital units. A total of 31 patients (50%) belonged to the ICU, while 14 (22.5%) came from the Pneumology Unit. Infectious diseases and Hematology wards both revealed seven patients (11.9%). Finally, two patients (3.2%) emerged from the Internal Medicine Unit, and one (1.6%) belonged to the Surgery Unit. The study documented 34 (54.8%) IPA patients and 28 (45.2%) non-IPA patients. Figure 1 illustrates these patients’ distribution within the analyzed hospital wards. Most IPA episodes derived from the Intensive Care Unit (46%), while 4% of the IPA cases emerged from the Infectious Diseases setting.

The study included 29 BAL, 22 bronchial aspirates, and 11 sputum samples. A total of 51 culture exams (82.2%) reported a positive result. As regards the IPA patients, *A. niger* (29.4%) and *A. fumigatus* (23.5%) were the most isolated species. *A. terreus* revealed a 20% incidence, while *A. flavus* reported 17.1%. Finally, one patient (2.8%) showed a mixed infection, reporting both *A. fumigatus* and *A. terreus* isolates. Among the same patients’ group, microscopy mainly reported positive results (97.1%). The galactomannan detection in IPA patients’ BAL samples accounted for 17 positive results (94.4%) with an index range of 1.23–12.16 (average galactomannan index value = 3.3) and 1 negative result (index = 0.0.3).

Overall, the PCR assay gathered 42 (67.7%) positive results, accounting for 100% among IPA patients. As regards non-IPA patients, 7 cases (25.9%) reported a molecular positive report. Table 1 summarizes general information and microbiological test results for both IPA and non-IPA patients.

As per patients’ characteristics, the antifungal treatment reported a correlation to the IPA condition (<0.05). Additionally, microbiological positive results (gathered through conventional, serological, and molecular assays) showed a significant correlation to the IPA condition (*p* < 0.05). Table 2 shows IPA patients’ risk factors, clinical conditions, and culture results.

Considering the detected fungal DNA load in IPA patients’ samples, the medium DNA quantity was 262,709 copies/mL (range 5–5,345,000 copies/mL). Among these molecular results, four cases revealed a medium DNA quantity of 41 copies/mL (range 5–78 copies/mL), classified as a load under the PCR significance level. A total of 11 non-IPA patients (17.7%) reported a positive PCR result, accounting for four cases (36.4%) with a DNA medium load of 592 copies/mL. On the other hand, seven non-IPA patients (63.6%) revealed a DNA quantity under the significance cut-off (average DNA quantity = 36 copies/mL). Figure 2 shows qPCR DNA loads (copies/mL) among IPA and non-IPA patients. Fungal DNA loads higher than 10^3^ in a logarithmic scale only regarded IPA patients, while non-aspergillosis episodes reported a maximum level of 10^3^ in the same measure’s scale. A statistical significance (*p* = 0.0004) emerged from these differences.

The authors produced a further distribution analysis (Figure 3), depending on the diagnosis (IPA or non-IPA) and the analyzed biological sample. The analysis demonstrated no statistical significance between fungal loads (copies/mL) and a specific respiratory sample both in IPA (*p* = 0.868) and non-IPA patients (*p* = 0.428).

Finally, Table 3 summarizes sensitivity, specificity, positive predictive value, and negative predictive value. PCR assay demonstrated the highest negative predictive value and sensitivity. Microscopy had the highest specificity and positive predictive value. Otherwise, low specificity rates emerged from culture and PCR.

## 4. Discussion

Pulmonary invasive aspergillosis is one of the most concerning fungal infections, recording high incidence rates among critically ill patients. *Aspergillus* spp. colonizes these patients’ airways, and its opportunistic attitude commonly leads to infection episodes due to different risk factors. Undoubtedly, the patient’s immunological condition, the prolonged ICU stay, or the use of corticosteroids contribute to possible *Aspergillus* spp. pulmonary infections after previous respiratory colonization [13,14,15]. Despite a low sensitivity rate and a long turn-around time, the conventional diagnosis still includes culture assays. Serological markers such as galactomannan efficiently support the probable aspergillosis diagnosis [16,17]. Diagnostic limitations lead to insufficient epidemiological reports and aspergillosis underestimation. Thus, innovative diagnostic methods integrated the conventional laboratory procedures. Published data described the real-time PCR for aspergillosis diagnosis in recent decades, reporting interesting sensitivity rates for qualitative and quantitative protocols [18,19].

Recent literature data confirmed the importance of performing quantitative molecular assays to support the invasive aspergillosis diagnosis [19]. According to this assumption, the present study evaluated the qPCR role in the laboratory aspergillosis diagnosis. We used an automated extraction and amplification system to detect *Aspergillus* spp. DNA on respiratory samples belonging to patients whose risk factors and clinical conditions were related to probable invasive pulmonary aspergillosis. First, the study confirmed the increasing prevalence of pulmonary aspergillosis episodes in the ICU settings [20]. In addition, our analysis enhanced the previously known local epidemiology of *Aspergillus* species diffusion, confirming the *Aspergillus fumigatus* predominance [21,22]. 

Our statistical analysis confirmed the elevated sensitivity of the BAL galactomannan detection in invasive pulmonary aspergillosis. Current guidelines require a minimum positivity cut-off value of 1. Consequently, we decided to reformulate the statistical analysis on BAL samples, documenting a sensitivity of 88.9%, a specificity of 100%, a positive predictive value of 100%, and a negative predictive value of 84.6%. Furthermore, we hypothesized a correlation between the BAL galactomannan index and the fungal DNA quantity. Specifically, all the patients who reported a negative galactomannan index (<0.5) had a PCR negative result (57.1%) or an average of 288 copies/mL non-significant fungal load (42.8%). As regards the BAL galactomannan index interval of 0.5-0-9, two patients (33.3%) had a PCR negative result, and four patients (66.6%) reported an average DNA quantity of 1500 copies/mL (10^3^ on a logarithmic scale). Finally, we observed an average fungal load of 504,510 copies/mL (10^5^ on a logarithmic scale), documenting a BAL galactomannan index higher than 1.

Despite these fundamental statistic confirmations, clinicians often find difficulties in collecting BAL samples, especially in non-intubated patients. On this premise, we aimed to evaluate the diagnostic value of other respiratory samples (sputum, bronchial aspirate). Current methodologies do not propose any standardization for galactomannan detection in these samples. One hypothesis regarded the PCR potential of these samples for the IPA diagnosis. Furthermore, we aimed to demonstrate a possible DNA quantity of various distributions depending on the colonization or infection condition. Our analysis confirmed these speculations, accounting for higher fungal DNA loads among IPA patients than non-IPA cases. Specifically, most of the IPA patients (88.5%) registered DNA quantity over the significance level (>120 copies/mL), while four patients (11.4%) reported no significant fungal DNA loads. As regards these four patients, only two cases (50%) showed a positive culture result (*A. niger* and *A. terreus*).

On the other hand, non-IPA patients mostly revealed negative PCR results (59.2%). The same patients’ group had 25.9% positive PCR with DNA quantity under the significance level, whereas only four patients (14.8%) overcame this cut-off, remaining under the 10^3^ in a logarithmic scale. Current guidelines do not standardize any cut-off in distinguishing colonization or infection conditions. However, the collected data suggest a future possibility to define a specific cut-off within a logarithmic scale, probably at the level of 10^3^ in a logarithmic scale. Unfortunately, our study did not determine this detail due to a reduced sample size and a single-centre involvement. Further studies shall include more samples and a multi-centre evaluation. Furthermore, some extended real-time PCR panels would be essential to distinguish the diverse *Aspergillus* species, which may correlate to different therapeutic choices and clinical outcomes. Antifungal resistance episodes, in fact, often report a species-specific attitude [23]. Scientific data already documented encouraging results in identifying *Aspergillus* species through molecular tests [19], but future directions may specifically regard metagenomics next-generation sequencing (mNGS) to optimize identification during diagnostic processes [19,24].

## 5. Conclusions

The study demonstrated that qPCR represents an interesting diagnostic tool, together with serological markers (galactomannan) and conventional methods (culture, microscopy). These methodologies support invasive pulmonary aspergillosis diagnosis. Despite this conclusion, further studies will be essential to extend preliminary speculations about fungal DNA quantity and colonization/infection conditions in high-risk patients.

## Figures and Tables

**Figure 1 microorganisms-13-00409-f001:**
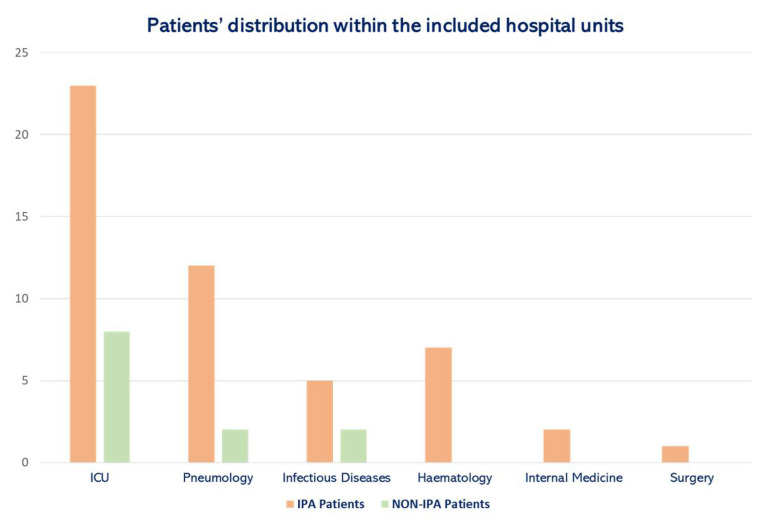
Patients’ distribution within the analyzed hospital wards. IPA Patients = Patients affected by probable invasive pulmonary aspergillosis; non-IPA Patients = Patients without probable invasive pulmonary aspergillosis; ICU = Intensive Care Unit.

**Figure 2 microorganisms-13-00409-f002:**
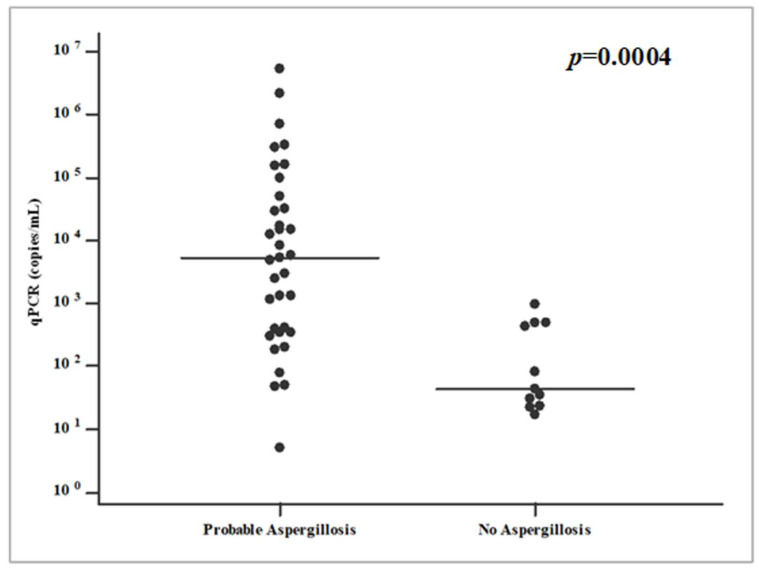
Fungal DNA loads (copies/mL) distribution among IPA and non-IPA patients in a logarithmic scale.

**Figure 3 microorganisms-13-00409-f003:**
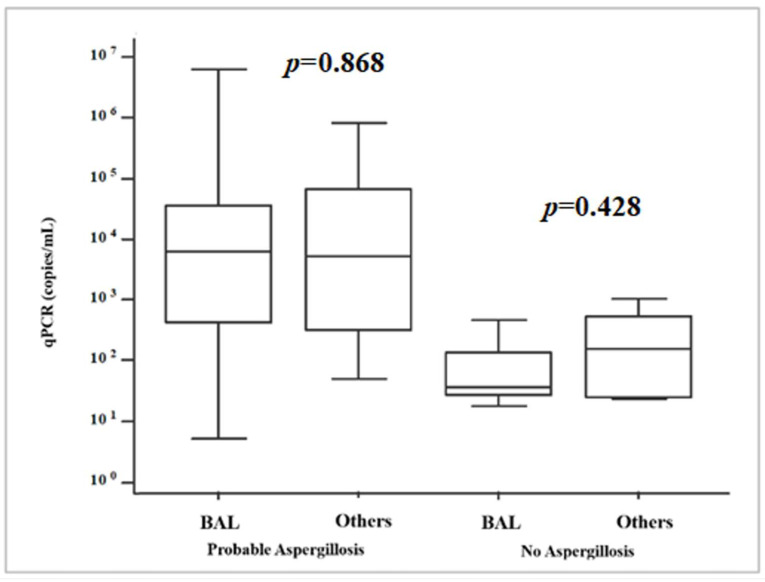
Fungal DNA loads (copies/mL) stratified depending on diagnosis (IPA or non-IPA) and sample type (BAL = bronchoalveolar lavage; Others = sputum, bronchial aspirate).

**Table 1 microorganisms-13-00409-t001:** General characteristics and microbiological data for the included patients’ respiratory samples.

Patients and Risk Factors	Total (*n* = 62)	Probable IPA (*n* = 35)	No IPA (*n* = 27)	*p* ^a^
Male sex (%)	30 (48.4)	15 (42.8)	9 (33.3)	0.448
Age, median years (range)	79 (19 to 85)	63 (19 to 85)	66 (29 to 84)	0.809
Antifungal treatment (no., %)	53 (85.5)	35 (100%)	19 (70.4)	<0.0001
**No. of patients (%) with a:**				
Positive microscopy ^a^	32 (51.6)	32 (91.4)	0	<0.0001
Positive culture	51 (82.2)	33 (94.3)	18 (66.7)	0.005
Positive GM ^b^	22 (75.9)	17 (94.4)	5 (45.4)	0.005
Positive PCR	46 (74.2)	35 (100)	11 (40.7)	<0.0001

^a^ Positive microscopy included filamentous fungi hyphal fragments or hyphae microscopic detection. ^b^ positive GM cases were calculated on the BAL samples (29). Abbreviations: IPA = Invasive pulmonary aspergillosis; GM = Galactomannan.

**Table 2 microorganisms-13-00409-t002:** IPA patients risk factors, clinical features, microbiological details, and therapeutic data.

No	Host Factors	Clinical Features	Respiratory Culture	GM ^a^	PCR ^b^	Antifungal Terapy
1	Chronic obstructive lung disease	Cavity	Negative	8.46	5	Itraconazole
2	Use of corticosteroids	Air crescent sign	*A. fumigatus*	12.16	2,175,083	Amphotericin B
3	COVID-19	Dense, well-circumscribed lesions	*A. terreus*	11.3	301,497	Amphotericin B
4	Use of corticosteroids	Nodule	*A. fumigatus*	5.7	155,877	Amphotericin B
7	Severe influenza	Air crescent sign	*A. niger*	3.4	5377	Voriconazole
8	Multiple myeloma	Dense, well-circumscribed lesions	*A. fumigatus*	6.9	8181	Amphotericin B
9	HSCT	Dense, well-circumscribed lesions	*A. niger*	-	48	Amphotericin B
10	Neutrophil deficiency	Air crescent sign	*A. niger*	-	713,875	Amphotericin B
11	Chronic obstructive lung disease	Cavity	*A. niger*	-	17,327	Caspofungin
12	Hematological malignancies	Dense, well-circumscribed lesions	*A. flavus*	-	330,791	Amphotericin B
14	Treatment with immunosuppressant	Tracheobronchial ulceration	*A. fumigatus*	1.3	302	Voriconazole
15	Solid organ transplant	Nodule	Negative	1.9	50	Caspofungin
17	Acute leukemias	Nodule	*A. terreus*	-	98,182	Amphotericin B
19	Acute leukemias	Nodule	*A. fumigatus*	-	2469	Amphotericin B
20	Glucocorticoid treatment	Cavity	*A. flavus*	-	162,433	Amphotericin B
25	Chronic obstructive lung disease	Cavity	*A. fumigatus*	-	181	Caspofungin
28	Chronic obstructive lung disease	Cavity	*A. fumigatus*	-	1167	Voriconazole
29	Acute leukemias	Air crescent sign	*A. terreus*	2.55	1295	Amphotericin B
31	COVID-19	Dense, well-circumscribed lesions	*A. niger*	9.49	2937	Amphotericin B
34	Treatment with immunosuppressant	Chronic obstructive lung disease	*A. terreus*	2.44	14,560	Caspofungin
35	Chronic obstructive lung disease	Cavity	*A. flavus*	-	345	Caspofungin
36	Acute leukemias	Air crescent sign	*A. terreus*	-	12,356	Caspofungin
37	Acute leukemias	Dense, well-circumscribed lesions	*A. terreus*	-	4765	Amphotericin B
38	Chronic obstructive lung disease	Cavity	*A. terreus*	-	78	Caspofungin
41	Acute leukemias	Air crescent sign	*A. fumigatus*	-	15,185	Amphotericin B
42	Chronic obstructive lung disease	Cavity	*A. flavus*	-	1295	Caspofungin
43	Hematological malignancies	Dense, well-circumscribed lesions	*A. flavus*	1.33	29,271	Amphotericin B
44	Chronic obstructive lung disease	Cavity	*A. niger*	0.84	5872	Itraconazole
45	Severe influenza	Air crescent sign	*A. fumigatus*	2.72	391	Caspofungin
46	Treatment with immunosuppressant	Dense, well-circumscribed lesions	*A. flavus*	-	197	Amphotericin B
47	Severe influenza	Dense, well-circumscribed lesions	*A. terreus*	-	50,459	Caspofungin
48	Chronic obstructive lung disease	Cavity	*A. niger*	0.03	408	Caspofungin
49	COVID-19	Dense, well-circumscribed lesions	*A. fumigatus*	6.34	5,345,000	Amphotericin B
50	Severe influenza	Air crescent sign	*A. fumigatus*	3.56	32,000	Amphotericin B
51	Chronic obstructive lung disease	Cavity	*A. fumigatus*	2.2	345	Voriconazole

^a^ GM in BAL. ^b^ copies/mL. Abbreviations: GM = galactomannan; PCR = Polymerase chain reaction.

**Table 3 microorganisms-13-00409-t003:** Sensitivity, specificity, positive predictive value, and negative predictive value for all the tested respiratory samples.

Techniques	Sensitivity (%)	Specificity (%)	PPV (%)	NPV (%)
Microscopy	91.4	100	100	90
Culture	94.3	32.1	64.7	81.8
GM *	94.4	54.5	77.3	85.6
PCR	100	59.2	76.1	100

Abbreviations: PPV = Positive predictive value; NPV = Negative predictive value. * Calculated on the total number of BAL samples (29).

## Data Availability

The authors described all the collected data within the manuscript.

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
