# Peer review of "The Role of Quantitative Real-Time PCR in the Invasive Pulmonary Aspergillosis Diagnosis: A Retrospective Study"

_microorganisms, 2025, doi:10.3390/microorganisms13020409_

Round 1
Reviewer 1 Report
Comments and Suggestions for Authors
In the manuscript “The role of quantitative Real-Time PCR in the diagnosis of invasive pulmonary aspergillosis: a retrospective study”, the authors propose a retrospective analysis of respiratory samples from patients with or without probable pulmonary aspergillosis, who underwent quantitative real-time PCR.
The aim of this work was to corroborate whether real-time PCR was sufficient to carry out the diagnosis of pulmonary aspergillosis, however, its results showed that this technique is insufficient to make the diagnosis, since it is essential to use other tools such as serological markers (galactomannan) and conventional methods (culture, microscopy).
It is an interesting and well-planned work, I only have one suggestion.
The authors should include in the discussion that real-time PCR is an important tool in the diagnosis of aspergillosis, as long as a specific probe is used, and better yet, one that identifies each species, since accurate identification of the species is important to provide adequate treatment, due to the increase in resistance to antifungals depending on the species. In addition, they should include in their discussion that the trend in the diagnosis of aspergillosis is the use of Metagenomics Next Generation Sequencing (mNGS) since it can identify the species of these fungi.
Standardize the references according to the format of the journal.
In table 1 correct “Positive Colture” to “Positive culture”
Author Response
In the manuscript “The role of quantitative Real-Time PCR in the diagnosis of invasive pulmonary aspergillosis: a retrospective study”, the authors propose a retrospective analysis of respiratory samples from patients with or without probable pulmonary aspergillosis, who underwent quantitative real-time PCR. The aim of this work was to corroborate whether real-time PCR was sufficient to carry out the diagnosis of pulmonary aspergillosis, however, its results showed that this technique is insufficient to make the diagnosis, since it is essential to use other tools such as serological markers (galactomannan) and conventional methods (culture, microscopy). It is an interesting and well-planned work, I only have one suggestion:
- Comment: The authors should include in the discussion that real-time PCR is an important tool in the diagnosis of aspergillosis, as long as a specific probe is used, and better yet, one that identifies each species, since accurate identification of the species is important to provide adequate treatment, due to the increase in resistance to antifungals depending on the species. In addition, they should include in their discussion that the trend in the diagnosis of aspergillosis is the use of Metagenomics Next Generation Sequencing (mNGS) since it can identify the species of these fungi.
Answer: Thank you for the interesting suggestion. The final part of the discussion section has been revised according to your advices. - Comment: Standardize the references according to the format of the journal.
Answer: The references have been revised. - Comment: In table 1 correct “Positive Colture” to “Positive culture”
Answer: It has been corrected.
Please, find all the requested changes highlighted in yellow.
Reviewer 2 Report
Comments and Suggestions for Authors
Thank you for inviting me to review this manuscript. It is interesting and well-written. I have some comments that could be of use:
1. Line 8: ‘critically ill patients’ instead of ‘critical patients’
2. English needs to be revised
3. Line 33: ‘transplantation’ instead of ‘transplants’
4. Line 38: ‘according to a scientific report’: I cannot see a reference for this
5. Line 83: Please state the country the study was conducted, right after the name of the hospital
6. Table 2: Please change amphotericin to amphotericin B
7. Table 1: The authors could add a line showing the PCR result. It would be interesting to provide a statistical comparison of the levels of copies/ml in the two different patient groups with numbers in the table. Graph 3 or 4 could be moved to the supplementary material in that case
8. Change ‘Graph’ to ‘Figure’
9. The authors show Graphs 1, 3, 4. I can’t find Graph 2
10. Table 3: Maybe the authors should add in the title of the table the information of where these samples were taken. If they are not all from the BAL, I think it should be stated. For example: ‘Sensitivity, specificity, positive predictive value, and negative predictive value of PCR in all respiratory samples’
11. I am skeptical about the cases where A. niger was identified. This species are commonly contaminants. On the other hand, many patients received treatment that is not first line (like echinocandins) or may not even have activity against molds (like fluconazole). Thus, I believe that the authors should provide data regarding the outcome of the patients. For example, patient 11 had a positive PCR with thousands copies/ml but with A. niger and was treated with fluconazole. Was this patient considered to have IPA? Because his whole story suggests not (unless he died)
12. I suggest the authors add a limitations subsection right at the end of the discussion section and before the conclusions. Some limitations for example have to do with the small sample size, or the fact that this is a single-center study
Comments on the Quality of English LanguageEnglish revision is suggested
Author Response
Thank you for inviting me to review this manuscript. It is interesting and well-written. I have some comments that could be of use:
- Comment: Line 8: ‘critically ill patients’ instead of ‘critical patients’
Answer: It has been corrected. - Comment: English needs to be revised
Answer: We revised the English language within the manuscript. - Comment: Line 33: ‘transplantation’ instead of ‘transplants’
Answer: It has been corrected. - Comment: Line 38: ‘according to a scientific report’: I cannot see a reference for this.
Answer: We apologize for the typo. It has been added. - Comment: Line 83: Please state the country the study was conducted, right after the name of the hospital.
Answer: “Italy” has been added. - Comment: Table 2: Please change amphotericin to amphotericin B.
Answer: We apologize for the typo. All the “amphotericin” expressions have been changed to “amphotericin B”. - Comment: Table 1: The authors could add a line showing the PCR result. It would be interesting to provide a statistical comparison of the levels of copies/ml in the two different patient groups with numbers in the table. Graph 3 or 4 could be moved to the supplementary material in that case.
Answer: Thank you for the observation. We decided to apply the Mann-Whitney U-test to elaborate a statistical evaluation between IPA and non-IPA patients about copies/ml. We reported a sentence to emphasize this detail (Lines 243-244). Furthermore, we elaborated statistical details depending on the analyzed clinical samples, remarking Graph 3 and 4. We decided to maintain these figures due to their direct contribution to the results’ description. - Comment: Change ‘Graph’ to ‘Figure’.
Answer: It has been corrected. - Comment: The authors show Graphs 1, 3, 4. I can’t find Graph 2.
Answer: We apologize for the typos. Figure 1,2, and 3 have been numbered again. We only produced 3 figures. - Comment: Table 3: Maybe the authors should add in the title of the table the information of where these samples were taken. If they are not all from the BAL, I think it should be stated. For example: ‘Sensitivity, specificity, positive predictive value, and negative predictive value of PCR in all respiratory samples.
Answer: Thank you for the observation. That information regarded all the respiratory samples; thus, we decided to revise the title. However, the galactomannan (GM) evaluation was obviously calculated on BAL samples and we specified this detail within the footnote. - Comment: I am skeptical about the cases where A. niger was identified. This species are commonly contaminants. On the other hand, many patients received treatment that is not first line (like echinocandins) or may not even have activity against molds (like fluconazole). Thus, I believe that the authors should provide data regarding the outcome of the patients. For example, patient 11 had a positive PCR with thousands copies/ml but with A. nigerand was treated with fluconazole. Was this patient considered to have IPA? Because his whole story suggests not (unless he died).
Answer: We understand the skepticism about the A. niger isolation. However, the EORTC guidelines established that Aspergillus spp. isolation along with clinical features (cavity) account for microbiological and clinical criteria defining the IPA condition.
Consequently, we classified some patients as IPA patients according to those guidelines even in the case of A. niger isolation. However, we consulted the clinical data again and noticed that patients with Aspergillus spp. positive culture and molecular result were initially under fluconazole regimen. However, the antifungal treatment switched to caspofungin after the microbiological data communication. According to this premise, we modified “fluconazole” in “caspofungin” for the above-mentioned patients (Patient 11 with A. niger and Patient 35 with A. flavus).
- Comment: I suggest the authors add a limitations subsection right at the end of the discussion section and before the conclusions. Some limitations for example have to do with the small sample size, or the fact that this is a single-center study.
Answer: Thank you for the suggestions. We added some sentences at the end of the discussion section to specify some limitations such as sample size, single-center study and other information on species identification requested by the other reviewer.
Please, find all the requested changes highlighted in yellow.
Reviewer 3 Report
Comments and Suggestions for Authors
Dear Authors,
I have reviewed the manuscript entitled "The role of quantitative Real-Time PCR in the invasive pulmonary aspergillosis diagnosis: a retrospective study" and found it well-written and of interest to the scientific community as it presents the potential use of qPCR as a rapid diagnostic method for invasive pulmonary aspergillosis. This could highly impact the management of infections caused by Aspergillus spp. and contribute to a better eradication rate of IPA.
The research data is well-organized and presented clearly, therefore I have only a few small observations (mainly spelling errors) to be considered, that are mentioned later:
1. Line 10: Please correct "represent" to "represents".
2. Line 15: Please correct "patients' with..." to "patients with".
3. Line 43: Please correct "Infections" to "infections".
4. Line 130: I recommend using initial capital letters for "Internal Medicine Unit" like for the other hospital units mentioned in the paragraph.
5. Line 144: I believe "isolation" should be replaced with "isolates" (please verify and correct if appropriate.
6. Tables 1 and 3: Please correct "Colture" to "Culture".
Author Response
Dear Authors,
I have reviewed the manuscript entitled "The role of quantitative Real-Time PCR in the invasive pulmonary aspergillosis diagnosis: a retrospective study" and found it well-written and of interest to the scientific community as it presents the potential use of qPCR as a rapid diagnostic method for invasive pulmonary aspergillosis. This could highly impact the management of infections caused by Aspergillus spp. and contribute to a better eradication rate of IPA. The research data is well-organized and presented clearly, therefore I have only a few small observations (mainly spelling errors) to be considered, that are mentioned later:
- Comment: Line 10: Please correct "represent" to "represents".
Answer: It has been corrected. - Comment: Line 15: Please correct "patients' with..." to "patients with".
Answer: It has been corrected. - Comment: Line 43: Please correct "Infections" to "infections".
Answer: It has been corrected. - Comment: Line 130: I recommend using initial capital letters for "Internal Medicine Unit" like for the other hospital units mentioned in the paragraph.
Answer: It has been corrected. - Comment: Line 144: I believe "isolation" should be replaced with "isolates" (please verify and correct if appropriate).
Answer: It has been corrected. - Comment: Tables 1 and 3: Please correct "Colture" to "Culture".
Answer: It has been corrected.
Please find all the requested changes highlighted in yellow.
Round 2
Reviewer 2 Report
Comments and Suggestions for Authors
The manuscript has been improved.